# Mechanisms Underlying the Stimulation of DUSP10/MKP5 Expression in Chondrocytes by High Molecular Weight Hyaluronic Acid

**DOI:** 10.3390/biomedicines13020376

**Published:** 2025-02-05

**Authors:** Wataru Ariyoshi, Jun Takeuchi, Sho Mitsugi, Ayaka Koga, Yoshie Nagai-Yoshioka, Ryota Yamasaki

**Affiliations:** 1Division of Infections and Molecular Biology, Department of Health Promotion, Kyushu Dental University, Fukuoka 803-8580, Japan; r20koga@fa.kyu-dent.ac.jp (A.K.); r16yoshioka@fa.kyu-dent.ac.jp (Y.N.-Y.); r18yamasaki@fa.kyu-dent.ac.jp (R.Y.); 2Medical Affairs, Seikagaku Corporation, Tokyo 100-0005, Japan; jun.takeuchi@seikagaku.co.jp; 3Second Department of Oral and Maxillofacial Surgery, Osaka Dental University, Osaka 540-0008, Japan; syo3next@gmail.com; 4School of Oral Health Sciences, Faculty of Dentistry, Kyushu Dental University, Fukuoka 803-8580, Japan

**Keywords:** chondrocyte, hyaluronic acid, CD44, DUSP10/MKP5, PI3K/Akt pathway, micro-RNA, inflammation

## Abstract

**Background/Objectives**: Previously, we reported that high molecular weight hyaluronic acid (HMW-HA) exerts chondroprotective effects by enhancing dual specificity protein phosphatase 10/mitogen-activated protein kinase (MAPK) phosphatase 5 (DUSP10/MKP5) expression and suppressing inflammatory cytokine-induced matrix metalloproteinase-13 (MMP13) expression in a human immortalized chondrocyte line (C28/I2 cells) via inhibition of MAPKs. The aim of this study was to elucidate the molecular mechanisms underlying the enhancement of DUSP10/MKP5 expression by HMW-HA in C28/I2 cells. **Methods**: C28/I2 cells were treated with HMW-HA, and the activation of intracellular signaling molecules was determined using Western blot analysis. The expression levels of mRNAs and microRNAs (miRNAs) were evaluated through real-time quantitative reverse transcription PCR analysis. **Results**: HMW-HA treatment induced Akt phosphorylation via interaction with CD44, and pretreatment with specific inhibitors of phosphatidylinositol-3 kinase/protein kinase B (PI3K/Akt) signaling attenuated the HMW-HA-induced expression of DUSP10/MKP5. HMW-HA suppressed the expression of miR-92a, miR-181a, and miR-181d. Loss-of-function and gain-of-function analyses of these miRNAs indicate that miR-92a, miR-181a, and miR-181d negatively regulate DUSP10/MKP5 expression. Moreover, HMW-HA-induced Akt phosphorylation was partially suppressed by miR-181a and miR-181d mimics. Finally, we found that HMW-HA activates RhoA-associated protein kinase (ROK) signaling, which contributes to Akt phosphorylation. **Conclusions**: These findings suggest that the induction of DUSP10/MKP5 expression by HMW-HA binding to CD44, leading to MMP13 suppression, involves multiple regulatory mechanisms, including PI3K/Akt and RhoA-activated ROK signaling, in addition to miRNA-mediated regulation. Elucidating these detailed molecular mechanisms may reveal novel biological activities that contribute to the therapeutic efficacy of HMW-HA against osteoarthritis.

## 1. Introduction

The intercellular spaces of articular cartilage are filled with aggrecan, a proteoglycan with diverse functional properties. Aggrecan maintains cartilage and other joint tissues by forming a matrix with collagen and hyaluronic acid (HA) [1]. HA is a non-sulfated linear glycosaminoglycan composed of repeating disaccharides of glucuronic acid and *N*-acetylglucosamine, and it is ubiquitously present in all tissues [2]. Increasing evidence indicates that HA, as a major component of the extracellular matrix in living organisms, contributes not only to the structuring and stability of the matrix [3] but also to various cellular functions such as proliferation and differentiation [4]. Joint inflammation associated with rheumatoid arthritis (RA) [5], osteoarthritis (OA) [6], and age-related degenerative changes [7] can lead to alterations in the localization, quantity, and quality of HA.

CD44, a type I membrane glycoprotein, is ubiquitously expressed in a variety of cells [8] and possesses an extracellular HA-binding domain [9]. Its intracellular domain is involved in intracellular signaling and interacts with cytoskeletal proteins [10]. The interaction of HA with CD44 has been implicated in the development and function of various tissues, including the skin, eye, neural tissue, immune system, and reproductive system. In joint tissues, signaling through CD44 in response to HA of various molecular weights regulates chondrocyte survival and apoptosis [8].

HA is commonly used in the treatment and management of severe arthritis, including OA and RA. In addition to its lubricating effects and viscoelasticity [11], growing evidence supports the pharmacological effects of intra-articular administration of HA, including anti-inflammatory [12], immunomodulatory [13], and analgesic effects [14]. We have previously identified multiple regulatory roles of HA in bone and cartilage metabolism [15,16]. Our earlier study demonstrated that high molecular weight HA (HMW-HA; 900 kDa) negatively regulates matrix metalloproteinase 13 (MMP13) expression induced by inflammatory cytokines through interaction with CD44 on human chondrocyte-like cells. This occurs via inhibition of mitogen-activated protein kinase (MAPK) activation, facilitated by enhanced expression of dual specificity protein phosphatase 10/MAPK phosphatase 5 (DUSP10/MKP5) [17].

DUSP family proteins inactivate target kinases by dephosphorylating serine/threonine and tyrosine residues [18]. MKPs, a subgroup of the DUSP family, negatively regulate the MAPK family, which includes extracellular signal-regulated kinase, c-Jun N-terminal kinase (JNK), and p38 MAPK [19]. Different MKP members exhibit varying substrate specificities for MAPKs, tissue distributions, subcellular localizations, and mechanisms of expression induction by extracellular stimuli. DUSP10/MKP5, which is uniformly present in the cytoplasm and nucleus, binds to and inactivates p38 MAPK and JNK, but not extracellular signal-regulated kinase [20].

While the role of DUSP10/MKP5 in the suppression of MMP13 expression by HMW-HA has been established, the detailed molecular mechanisms underlying the regulation of DUSP10/MKP5 expression through its interaction with CD44 remain unclear. The aim of this study was to elucidate the molecular mechanisms of DUSP10/MKP5 induction by HMW-HA treatment in chondrocytes, focusing on PI3K/Akt-mediated signaling and microRNA (miRNA) expression regulation.

## 2. Materials and Methods

### 2.1. Reagents and Antibodies

HMW-HA (ARTZ^®^, molecular weight: 900 kDa) was supplied by Seikagaku Corporation (Tokyo, Japan). CD44 function-blocking monoclonal antibody (Hermes-1), LY294002, simvastatin, Y-27632, and anti-β-actin monoclonal antibody (AC-15) were purchased from Merck Millipore (Darmstadt, Germany). Anti-phosho-Akt (Ser473) monoclonal (D9E), anti-Akt polyclonal, and anti-DUSP10/MKP5 polyclonal antibodies were obtained from Cell Signaling Technology (Beverly, MA, USA). Rabbit IgG HRP-linked whole and mouse IgG HRP-linked whole antibodies were obtained from Amersham™ GE Healthcare (Amersham, UK). Recombinant human TNF-α was purchased from R&D Systems (Minneapolis, MN, USA).

### 2.2. Cell Culture

The immortalized human chondrocyte cell line C28/I2 [21,22] was obtained from Merck Millipore and maintained in Dulbecco’s modified Eagle’s medium (DMEM; FUJIFILM Wako Pure Chemical Co., Osaka, Japan) containing 10% fetal bovine serum (FBS; Merck Millipore, Darmstadt, Germany), 100 U/mL penicillin G, and 100 µg/mL streptomycin (Wako Pure Chemical Industries, Osaka, Japan) at 37 °C with 5% CO_2_. The cells were stimulated with HMW-HA (100 µg/mL) for the indicated times. In some experiments, the cells were pretreated with CD44 mAb (20 µg/mL) for 2 h, LY294002 (10 µM) for 1 h, simvastatin (10 µM) for 24 h, or Y27632 (10 µM) for 24 h prior to stimulation with HMW-HA.

### 2.3. Western Blotting

Whole-cell lysates were extracted from cultured C28/I2 cells, and total protein concentrations were determined according to previously reported protocol [23]. Extracted proteins were subjected to Western blot analysis based on a previously established protocol [24], using primary antibodies against phospho-Akt, Akt, DUSP10/MKP5, and β-actin, followed by detection with the GelDoc™ XR Plus digital system (version 2.0.1; Bio-Rad, Hercules, SA, USA). The relative band intensities were calculated by normalization to β-actin or Akt intensity in the same sample.

### 2.4. Real-Time Quantitative Reverse Transcription PCR (RT-qPCR)

The total RNA was isolated from cultured C28/I2 cells and reverse transcribed into cDNA according to previously reported protocol [23]. PCR products were detected using a qPCR Brilliant III SYBR Master Mix with ROX (Agilent Technologies, Inc., Santa Clara, CA, USA). The following primer sequences were used: GAPDH, 5′-GACGGCCGCATCTTCTTGA-3′ (forward) and 5′-CACACACCGACCTTCACCATTTT-3′ (reverse); DUSP10/MKP5, 5′-CCAAGGAGCTGTCCACATTA-3′ (forward) and 5′-GCCTTCCCTACAGGAAATCAA-3′ (reverse); and MMP13, 5′-TGTTGCGCATGAGTTC-3′ (forward) and 5′-TGCTCCAGGGTCCTTGGA-3′ (reverse). Thermal cycling and fluorescence detection were performed using the AriaMx Real-Time PCR system (version 1.6; Agilent Technologies). Relative mRNA expression levels were calculated using the 2^−∆∆Ct^ method with GAPDH as the internal reference.

### 2.5. RT-qPCR for miRNA

Total RNA extracted from cultured C28/I2 cells was reverse transcribed into cDNA using the miRCURY LNATM RT Kit (QIAGEN, Valencia, CA, USA) according to the manufacturer’s instructions. The reverse transcription reactions were incubated at 42 °C for 60 min followed by inactivation at 95 °C for 5 min using a Thermal Cycler GeneAtlas (ASTEC, Tokyo, Japan). RT-qPCR amplification was performed on an AriaMx Real-Time PCR system using the miRCURY LNATM SYBR^®^ Green PCR Master Mix (QIAGEN). The following cycling conditions were used: 95 °C for 120 s, followed by 40 cycles of 95 °C for 10 s and 56 °C for 30 s. Relative miRNA expression levels were calculated with the 2^–∆∆Ct^ method with U6 as the internal reference.

### 2.6. Transfection with miRNA Inhibitors

miRCURY LNA^TM^ miRNA Power Inhibitors (QIAGEN, Valencia, CA, USA) were used to suppress the effects of miR-92a, miR-181a, and miR-181d. C28/I2 cells (2.4 × 10^6^ cells) were transfected with each inhibitor (150 pmol) using Lipofectamine^®^ 3000 Reagent (Thermo Fisher Scientific, Waltham, MA, USA) according to the manufacturer’s instructions. miRCURY LNA^TM^ miRNA Power Inhibitor Control (QIAGEN, Valencia, CA, USA) served as a negative control and consisted of a sequence with no significant homology (>70%) to any sequence from any organism in the NCBI and miRBase databases. Transfected cells were cultured in 6-well plates for 24 h prior to stimulation with HMW-HA and TNF-α.

### 2.7. Transfection with miRNA Mimics

miRCURY LNA^TM^ miRNA Mimics (QIAGEN, Valencia, CA, USA) were used to enhance the effects of miR-92a, miR-181a, and miR-181d. The miRNA mimics (0.1 nM) for miR-92a (AGGUUGGGAUCGGUUGCAAUGCU), miR-181a (AACAUUCAACGCUGUCGGUGAGU), and miR-181d (AACAUUCAUUGUUGUCGGUGGGU) were introduced into the C28/I2 cells (1.0 × 10^6^ cells) using an NEPA21 Super Electroporator (Nepa Gene Co., Ltd., Chiba, Japan). Two pulses with a voltage of 125 V/20 V with a width of 5 ms/50 ms were applied. Transfected cells were immediately suspended in prewarmed DMEM supplemented with 10% FBS and cultured in 6-well plates for 24 h prior to stimulation with HMW-HA.

### 2.8. Pull-Down Assay for Rho Activity

The amount of active RhoA was measured using a pull-down assay with Rho Assay Reagent (Merck Millipore, Darmstadt, Germany) according to the manufacturer’s instructions. Briefly, treated C28/I2 cells were washed twice with ice-cold Tris-buffered saline and lysed in Mg^2+^ Lysis/Wash Buffer (Merck Millipore, Darmstadt, Germany) containing a protease and phosphatase inhibitor cocktail (Thermo Fisher Scientific, Waltham, MA, USA). The lysates were clarified by centrifugation at 14,000× *g* for 5 min. The supernatant was incubated with Rho Assay Reagent bead slurry for 45 min with gentle agitation at 4 °C. The beads were then washed three times with Mg^2+^ Lysis/Wash Buffer and re-suspended in 500 µL of Laemmli reducing sample buffer (Bio-Rad, Hercules, SA, USA). The precipitated GTP-bound RhoA proteins were then analyzed via Western blot using an anti-Rho (-A, -B, -C) monoclonal antibody (clone 55; Merck Millipore, Darmstadt, Germany).

### 2.9. Statistical Analysis

Each experiment was performed at least three times. Data were analyzed using JMP^®^ software version 10.0.2 (SAS Institute Inc., Cary, NC, USA) and expressed as the mean ± standard deviation. To compare two groups, Student’s *t*-test was used. For comparisons among three or more groups, statistical differences were determined using one-way analysis of variance followed by Tukey’s or Dunnett’s post hoc test. Statistical significance was set at *p* < 0.05.

## 3. Results

### 3.1. Interaction of HMW-HA with CD44 Enhances DUSP10/MKP5 Expression via Activation of PI3K/Akt Signaling

We first evaluated the effect of HMW-HA on DUSP10/MKP5 expression in C28/I2 cells. HMW-HA treatment stimulated DUSP10/MKP5 protein expression in a time-dependent manner (Figure 1A). Additionally, HMW-HA induced transient phosphorylation of Akt protein (Figure 1B). This phosphorylation was reduced to basal levels when CD44 receptors were blocked using a function-blocking monoclonal antibody (Figure 1C). Furthermore, inhibition of the PI3K/Akt signaling pathway using a specific inhibitor (LY294002) attenuated HMW-induced mRNA DUSP10/MKP5 mRNA expression in C28/I2 cells (Figure 1D).

### 3.2. HMW-HA Inhibits the Biosynthesis of miR-92a, miR-181a, and miR-181d

Next, we examined the effect of HMW-HA on the biosynthesis of miRNAs known to regulate DUSP10/MKP5 gene expression. Administration of HMW-HA noticeably reduced the levels of miR-92a, miR-181a, and miR-181d in C28/I2 cells, whereas it had no effect on the biosynthesis of miR-181b (Figure 2A). The reduction in miR-92a, miR-181a, and miR-181d levels by HMW-HA was reversed upon pretreatment with a CD44 function-blocking antibody (Figure 2B).

### 3.3. Blockade of Hybridization Activity of miR-92a, miR-181a, and miR-181d Enhances DUSP10/MKP5 Expression

To investigate the regulatory effects of miRNAs modified by HMW-HA on DUSP10/MKP5 expression, miRCURY LNA^TM^ miRNA Power Inhibitors were used to suppress hybridization of target miRNAs with their interacting partners. Transfection with inhibitors for miR-92a and miR-181a, but not for miR-181d, significantly upregulated DUSP10/MKP5 mRNA expression (Figure 3A). Inhibitors for miR-92a, miR-181a, and miR-181d enhanced DUSP10/MKP5 protein levels in C28/I2 cells (Figure 3B). Furthermore, the miR-92a and miR-181a inhibitors downregulated MMP13 mRNA expression induced by TNF-α in C28/I2 cells, whereas the miR-181d inhibitor did not have a significant effect on this expression (Figure 3C).

### 3.4. Enhanced Activity of miR-92a, miR-181a, and miR-181d Suppresses DUSP10/MKP5 Expression

To increase the proportion of RNA-induced silencing complexes containing mR-92a, miR-181a, and miR-181d, we utilized miRCURY LNA^TM^ miRNA Mimics. Transfection with the miR-92a mimic significantly downregulated DUSP10/MKP5 mRNA expression in C28/I2 cells (Figure 4A). Furthermore, mimics for mR-92a, miR-181a, and miR-181d suppressed DUSP10/MKP5 protein expression (Figure 4B).

### 3.5. Enhancement of miR-181a and miR-181d Activity Inhibits HMW-HA-Induced Activation of PI3K/Akt Signaling

The interaction between PI3K/Akt-mediated signaling and miRNA expression, which plays a role in the upregulation of DUSP10/MKP5 expression by HMW-HA, was further investigated. Suppression of miR-92a, miR-181a, and miR-181d using miRCURY LNA^TM^ miRNA Power Inhibitors did not exhibit any significant effect on HMW-HA-induced Akt phosphorylation in C28/I2 cells (Figure 5A). Conversely, the miR-181a and miR-181d mimics significantly suppressed Akt phosphorylation induced by HMW-HA (Figure 5B).

### 3.6. RhoA and RhoA-Associated Protein Kinase (ROK) Inhibitors Attenuate PI3K/Akt Activation Mediated by HMW-HA

Finally, to examine the role of RhoA in PI3K/Akt activation in HMW-HA-stimulated C28/I2 cells, we evaluated RhoA activation in C28/I2 cells using a pull-down assay. HMW-HA treatment significantly increased the levels of GTP-bound RhoA, indicating RhoA activation (Figure 6A). Pretreatment with the Rho (simvastatin) or the ROK inhibitor (Y27632) effectively inhibited the phosphorylation of Akt induced by HMW-HA (Figure 6B).

## 4. Discussion

MMP13, a collagen type II-degrading enzyme, is strongly implicated in the pathogenesis of OA, as demonstrated by studies on knockout [25] and overexpressing [26] mouse models. It has been reported that p38 MAPK and JNK activation are involved in the induction of MMP13 expression in chondrocytes [27,28]. Furthermore, it has been demonstrated that the promoters of *MMP* gene harbor a binding region for AP-1 [29], a transcription factor activated by p38 MAPK [30] and JNK [31]. Hence, the induction of DUSP10/MKP5 expression in chondrocytes is suggested to be the main pathway for suppression of TNF-α-induced MMP13 expression by HMW-HA. Understanding the molecular mechanism by which DUSP10/MKP5-mediated suppression of MMP13 expression occurs via HMW-HA is crucial for developing more effective strategies to control the progression of OA. In a previous study, we found that hyaluronan oligosaccharides enhance Akt phosphorylation at the Thr308 and Ser473 residues in C28/I2 cells [32]. Furthermore, activation of the PI3K/Akt pathway by insulin has been shown to stabilize DUSP10/MKP5 in glioblastoma [33]. Consistent with these findings, this study demonstrated that HMW-HA, like hyaluronan oligosaccharides, promotes Akt phosphorylation in C28/I2 cells. Moreover, the use of selective inhibitors suggests that the PI3K/Akt-mediated pathway activated by HMW-HA is involved in the induction of DUSP10/MKP5 expression.

miRNAs are 21–25 nucleotide-long, single-stranded, non-coding RNAs that regulate gene expression in eukaryotic cells by binding to the 3′-untranslated region (UTR) of target genes. This binding leads to mRNA destabilization and translational repression, thereby suppressing protein production. miRNA-mediated transcriptional repression plays a crucial role in various cellular processes [34]. Numerous miRNAs have been implicated in the regulation of cartilage homeostasis [35] and the pathogenesis of OA [36,37] and RA [38]. Furthermore, intra-articular injection of HMW-HA in OA patients has been shown to increase miR-452-3p expression, which correlates with improved clinical scores [39].

Previous studies have reported that miR-92a is involved in the proliferation of human pancreatic cancer cells [40] and differentiation of T cells into the Th1 phenotype [41] via regulation of DUSP10/MKP5 expression. Similarly, in hepatocytes, miR-181a, miR-181b, and miR-181d have been shown to influence carcinogenesis by activating p38 MAPK through suppression of DUSP10/MKP5 expression [42]. In human chondrocytes, oligosaccharides of HA have been reported to regulate inflammatory responses via miR-21 [43] and miR-146a [44], but the involvement of microRNAs in the biological activity of HMW-HA is unclear. Our findings indicate that HMW-HA induces a transient repression of miR-92a, miR-181a, and miR-181d via interaction with CD44 in C28/I2 cells. In addition, analysis using the TargetScan 8.0 online database (http://www.targetscan.org/ (accessed on 12 July 2022)) predicted that miR-92a, miR-181a, and miR-181d directly bind to the 3′-UTR of DUP10/MKP5 mRNA (Table 1). These findings suggest that the suppression of miR-92a, miR-181a, and miR-181d by HMW-HA may be a key mechanism regulating DUSP10/MKP5 expression. To consolidate the results of the bioinformatics analysis, direct binding of candidate miRMAs to the UTR of DUSP10/MKP5 should be demonstrated using analytical methods such as luciferase assays.

HA has the ability to bind to membrane proteins such as CD44 (receptor for hyaluronan-mediated motility) and LYVE1 (lymphatic vessel endothelial hyaluronan receptor 1) [45]. The interaction of HMW-HA with CD44 has been shown to activate the PI3K/Akt signaling pathway in various cell types, including retinal progenitor cells [46], CD8 T cells [47], and dental pulp cells [48]. Furthermore, in tumor cells, this interaction regulates miRNA expression and function [49,50]. In this study, the use of neutralizing antibodies suggests that the binding of HMW-HA to the CD44 receptor on the surface of C28/I2 cells is required for PI3K/Akt activation and miRNA inhibition.

To further elucidate the role of miRNAs suppressed by HMW-HA in regulating DUSP10/MKP5 expression, we performed functional suppression experiments of target miRNAs using Power Inhibitors. These inhibitors are antisense oligonucleotides that bind tightly to target miRNAs in transfected cells, forming a double strand and inhibiting miRNA function. In C28/I2 cells transfected with inhibitors for miR-92a and miR-181a, DUSP10/MKP5 expression was enhanced, while TNF-α-induced MMP13 expression was suppressed. These findings suggest that the inhibition of miR-92a and miR-181a by HMW-HA is partially responsible for the suppression of TNF-α-induced MMP13 expression via inhibition of p38 MAPK and JNK activation by DUSP10/MKP5.

As a complementary approach, miRNA mimics were utilized to evaluate their functional impact. The innovative mimic design used in this study includes three RNA strands instead of the conventional two-strand structure. Interestingly, transfection with a miR-92a mimic significantly reduced both DUSP10/MKP*5* gene and protein levels. However, miR-181a and miR-181d mimics only significantly suppressed DUSP10/MKP5 protein expression. miRNAs have been reported to negatively regulate protein synthesis by directly suppressing translation in addition to mRNA degradation and destabilization [51]. Future studies are required to analyze the detailed mechanisms of DUSP10/MKP5 repression by miRNAs regulated by HMW-HA in C28/I2 cells.

Previous studies have reported that miR-92a [52,53,54], miR-181a [55], and miR-181d [56] inhibit the PI3K/Akt signaling pathway in various cells. Therefore, we investigated the ability of these miRNAs to modulate this pathway in C28/I2 cells. Akt phosphorylation induced by HMW-HA was moderately suppressed in cells transfected with miR-181a and miR-181d mimics, suggesting that suppression of these miRNAs may be involved in PI3K/Akt activation by HMW-HA. However, the effects were minor, and Akt phosphorylation levels remained unchanged in cells transfected with Power Inhibitors of these miRNAs.

Finally, we focused on ROK signaling, which is activated by the interaction of HMW-HA with CD44 in various cells, including keratinocytes [57], osteoblasts [15], and cancer cells [58]. In this study, the administration of HMW-HA significantly increased active GTP-bound RhoA levels, suggesting that interaction between HMW-HA and CD44 also induces RhoA signaling in C28/I2 cells. It has been reported that RhoA-mediated Akt activation is important for cell proliferation, metabolism, and function [59,60]. Simvastatin, a mevalonate synthesis inhibitor, and Y-27632, a pyridine derivative, inhibit ROCK signaling and have been widely used to explore the role of RhoA-activated ROCK signaling [61,62], indicating the involvement of this pathway in PI3K/Akt activation by HMW-HA.

As illustrated in Figure 7, the induction of DUSP10/MKP5 expression through the binding of HMW-HA to CD44, which leads to the suppression of MMP13 expression, involves multiple regulatory mechanisms, including PI3K/Akt and RhoA-activated ROK signaling, in addition to miRNAs. The reduced production of MMP13 via DUSP10/MKP5 expression may contribute to the rescue of inflammatory bone and cartilage destruction in OA joints. On the other hand, the PI3K/Akt/nuclear factor kappa B (NF-κB) signaling pathway has also been reported to be involved in triggering chondrocyte inflammation with increased MMP13 [63]. The modulatory potential of HMW-HA for the NF-κB-pathway in addition to MAPK is currently under investigation in our laboratory. The limitation of this study is its reliance on an in vitro model using cell lines, which may not capture the interplay of multiple factors regulating DUSP10/MKP5 expression. Administration of HMW-HA to rabbit and canine OA models induced by menisectomy or anterior cruciate ligament injury has been reported to have joint-protective biological effects [64,65]. Elucidation of the expression, localization, and interaction of these candidate molecules following HMW-HA administration in animal models may facilitate the development of novel therapeutic strategies leveraging HMW-HA for OA treatment. In addition, the expression levels of miRNAs and DUSP10/MKP5 and the activation levels of PI3K/Akt and RhoA-activated ROK signaling in HMW-HA-treated human articular chondrocytes should be investigated. Since DUSP10/MKP5 is involved in inflammatory regulation, the findings could have broader implications for other rheumatic diseases beyond cartilage degeneration, including those affecting synovial cells or the immune system. Furthermore, DUSP10/MKP5 has been shown to negatively modulate retroviral-induced inflammatory responses [66] and to induce macrophage polarization from M1 to M2 phenotype [67]. Targeting the molecular pathways identified in this study may lead to novel therapeutic interventions for a range of inflammatory disorders.

## Figures and Tables

**Figure 1 biomedicines-13-00376-f001:**
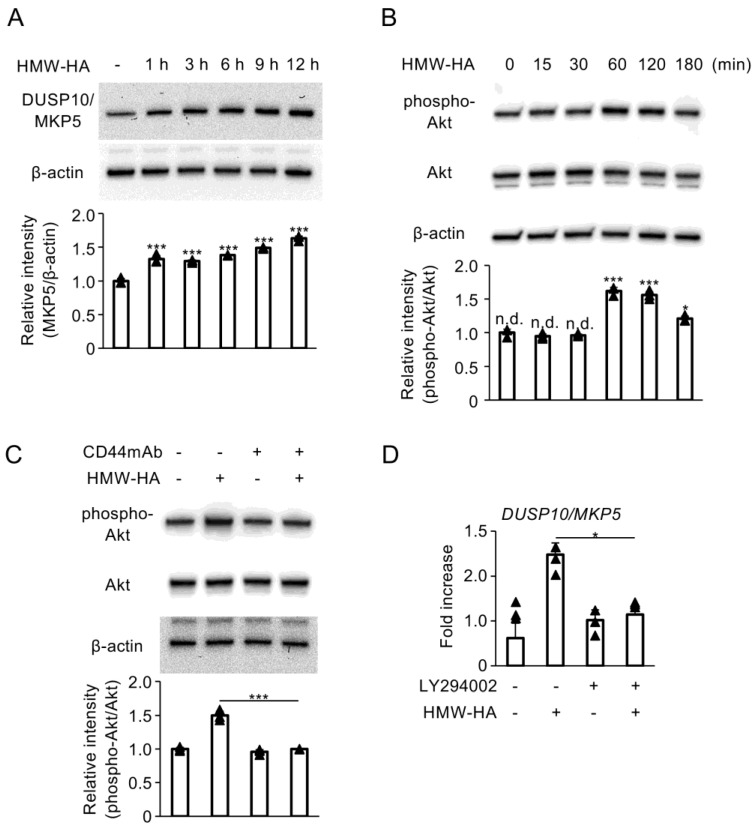
(**A**,**B**) C28/I2 cells were incubated with HMW-HA (100 µg/mL) for the indicated time periods. (**C**) C28/I2 cells were subjected to pretreatment with CD44mAb (20 µg/mL) for 2 h, followed by stimulation with HMW-HA (100 µg/mL) for 1 h. Whole cell lysates were subjected to SDS-PAGE, followed by Western blot analysis of DUSP10/MKP5, phospho-Akt, and Akt. Equivalent protein loading was confirmed by β-actin levels. Densitometric analyses of the respective blots are shown at the bottom. Bars represent the means ± standard deviation (SD) of relative band intensities normalized to β-actin (**A**) or Akt (**B**,**C**) from independent triplicate samples. (**D**) C28/I2 cells were pretreated with LY294002 (10 µM) for 1 h, followed by stimulation with HMW-HA (100 µg/mL) for 1 h. DUSP10/MKP5 mRNA expression was determined by real-time quantitative reverse transcription PCR (RT-qPCR). Data represent fold changes in mRNA copy numbers from independent triplicate samples. Data were analyzed using Dunnett’s test (**A**,**B**) or Tukey’s test (**C**,**D**) following one-way analysis of variance (ANOVA). * *p* < 0.05, *** *p* < 0.0001, and n.d., no difference, compared with untreated cells (**A**,**B**) or HMW-HA-treated cells (**C**,**D**).

**Figure 2 biomedicines-13-00376-f002:**
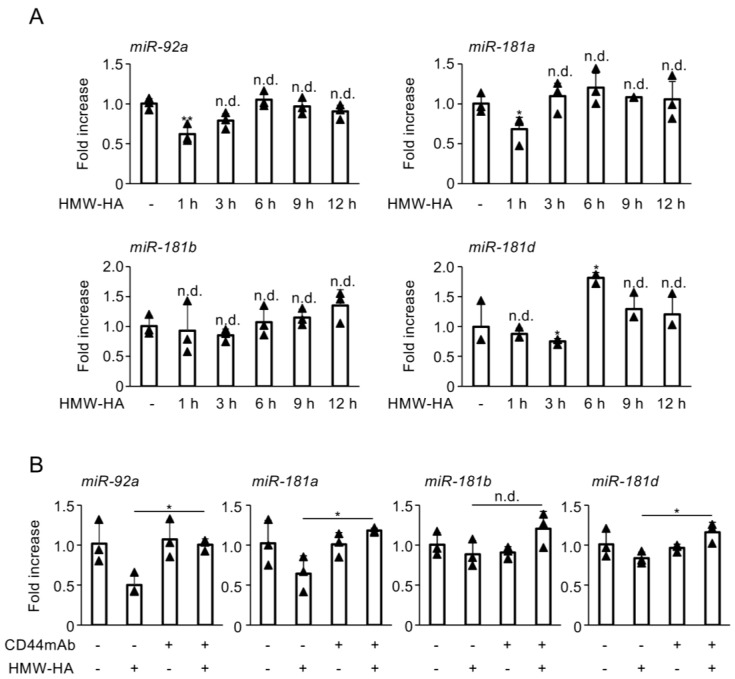
(**A**) C28/I2 cells were incubated with HMW-HA (100 µg/mL) for the indicated time periods. (**B**) C28/I2 cells were pretreated with CD44mAb (20 µg/mL) for 2 h, followed by stimulation with HMW-HA (100 µg/mL) for 1 h. MicroRNAs (miRNAs) miR-92a, miR-181a, miR-181b, and miR-181d expression was determined using RT-qPCR. Data represent fold changes in miRNA copy numbers from independent triplicate samples. Bars represent the means ± SD. Data were analyzed using Dunnett’s test (**A**) or Tukey’s test (**B**) following one-way analysis of variance (ANOVA). * *p* < 0.05, ** *p* < 0.01, and n.d., no difference, compared with untreated cells (**A**) or HMW-HA-treated cells (**B**).

**Figure 3 biomedicines-13-00376-f003:**
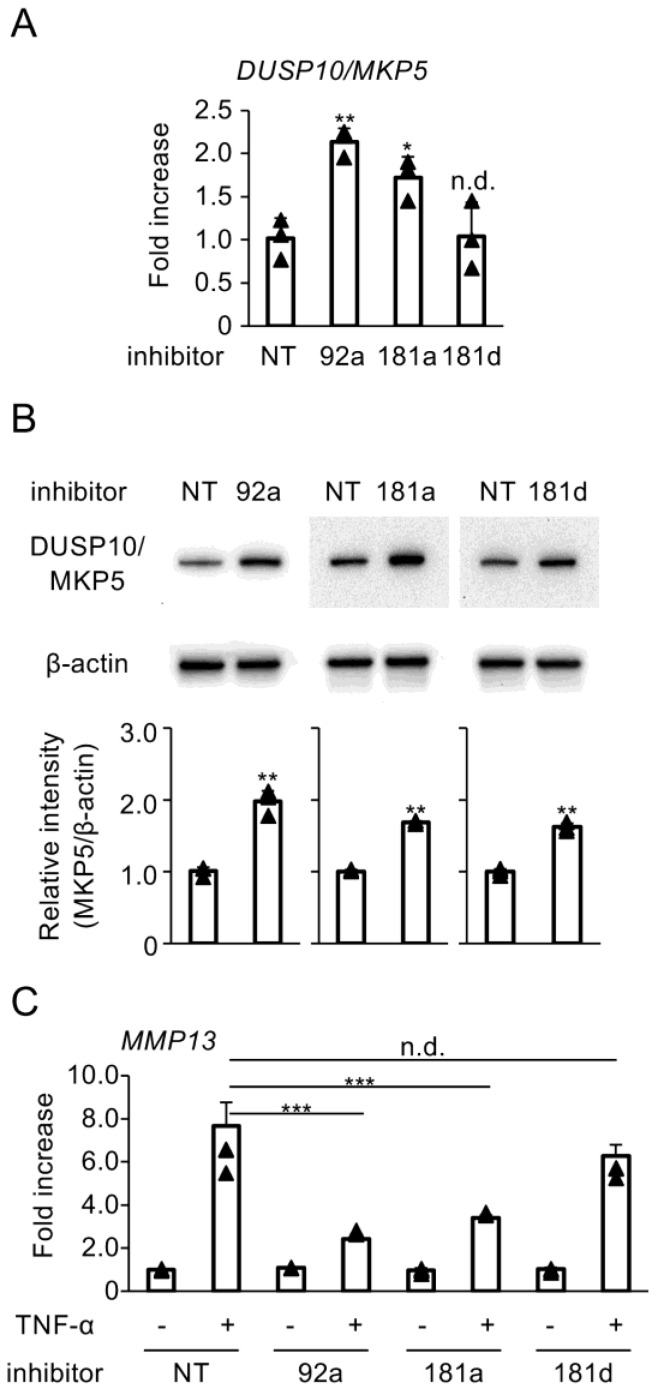
(**A**,**B**) C28/I2 cells were transfected with miRCURY LNA^TM^ miRNA Power Inhibitors for NT control (NT), miR-92a (92a), miR-181a (181a), or miR-181d (181d) and cultured for 24 h. (**A**) DUSP10/MKP5 mRNA expression was determined by RT-qPCR. Data represent fold changes in mRNA copy numbers from independent triplicate samples. (**B**) Whole cell lysates were subjected to SDS-PAGE and analyzed using Western blot for DUSP10/MKP5 expression. Equivalent protein loading was confirmed by β-actin levels. Densitometric analysis of the blot is shown at the bottom. Bars represent means ± SD of relative band intensities normalized to β-actin from independent triplicate samples. (**C**) C28/I2 cells were transfected with the same miRCURY LNA^TM^ miRNA Power Inhibitors and cultured for 24 h, followed by stimulation with TNF-α (20 ng/mL) for 6 h. *MMP13* mRNA expression was determined by RT-qPCR. Data represent fold changes in mRNA copy numbers from independent triplicate samples. Bars represent means ± SD. Data were analyzed using Dunnett’s test (**A**) or Tukey’s test (**C**) following one-way analysis of variance (ANOVA). For two-group comparisons, a two-sided Student’s *t*-test was used (**B**). * *p* < 0.05, ** *p* < 0.01, *** *p* < 0.0001, and n.d., no difference, compared with NT-transfected cells (**A**,**B**) or NT-transfected and TNF-α-treated cells (**C**).

**Figure 4 biomedicines-13-00376-f004:**
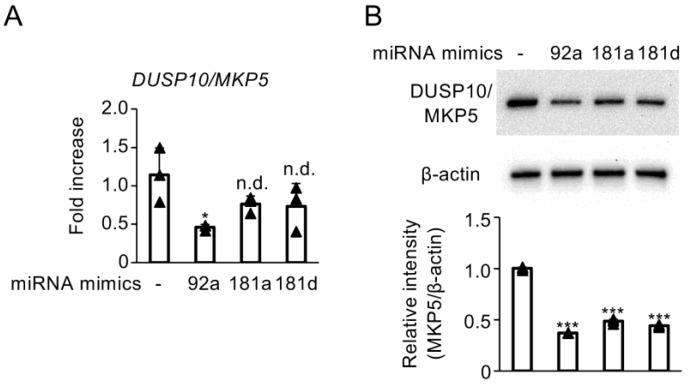
(**A**,**B**) C28/I2 cells were transfected with miRCURY LNA^TM^ miRNA Mimics for miR-92a (92a), miR-181a (181a), or miR-181d (181d) and cultured for 24 h. (**A**) DUSP10/MKP5 mRNA expression was determined by real-time RT-qPCR. Data represent fold changes in mRNA copy numbers for independent triplicate samples. (**B**) Whole cell lysates were subjected to SDS-PAGE and analyzed using Western blot for DUSP10/MKP5 expression. Equivalent protein loading was confirmed by β-actin levels. Densitometric analyses of the respective blots are shown at the bottom. Bars represent the means ± SD of relative band intensities normalized to β-actin from independent triplicate samples. Data were analyzed using Dunnett’s test following one-way analysis of variance (ANOVA). * *p* < 0.05, *** *p* < 0.0001, and n.d., no difference, compared with untreated cells.

**Figure 5 biomedicines-13-00376-f005:**
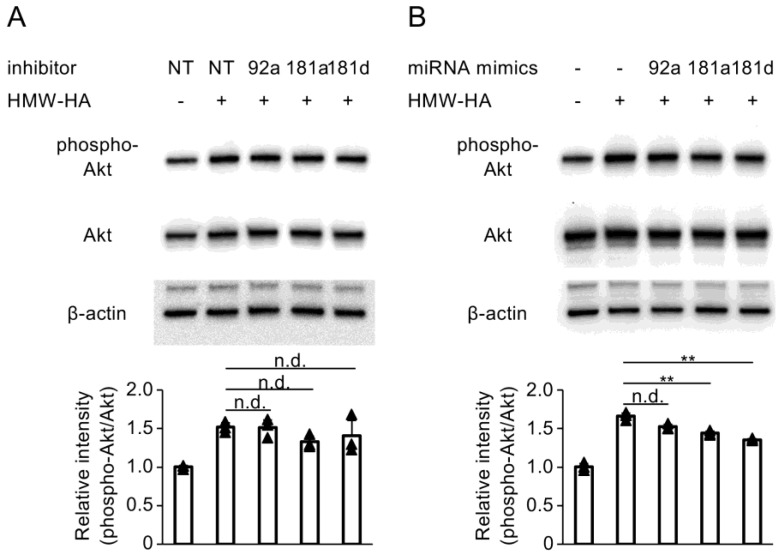
(**A**) C28/I2 cells were transfected with miRCURY LNA^TM^ miRNA Power Inhibitors for NT control (NT), miR-92a (92a), miR-181a (181a), or miR-181d (181d) and cultured for 24 h, followed by stimulation with HMW-HA (100 µg/mL) for 1 h. (**B**) C28/I2 cells were transfected with miRCURY LNA^TM^ miRNA Mimics for miR-92a (92a), miR-181a (181a), or miR-181d (181d) and cultured for 24 h, followed by stimulation with HMW-HA (100 µg/mL) for 1 h. Western blot analysis was performed to evaluate phospho-Akt and Akt expression. Equivalent protein loading was confirmed by β-actin levels. Densitometric analyses of the respective blots are shown at the bottom. Bars represent the means ± SD of relative band intensities normalized to Akt from independent triplicate samples. Data were analyzed using Tukey’s test following one-way analysis of variance (ANOVA). ** *p* < 0.01 and n.d., no difference, compared with the NT-transfected and HMW-HA-treated cells (**A**) or HMW-HA-treated cells (**B**).

**Figure 6 biomedicines-13-00376-f006:**
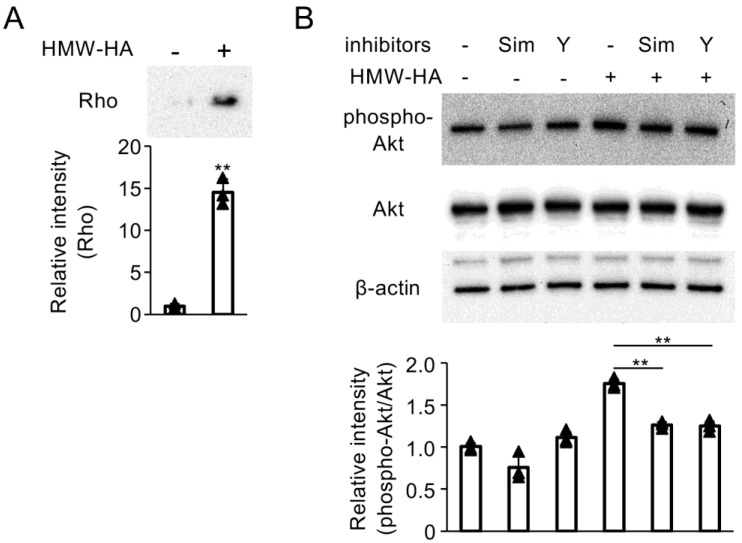
(**A**) C28/I2 cells were incubated with HMW-HA (100 µg/mL) for 30 min. Whole cell lysates were subjected to affinity precipitation using a Rho Activation Assay Kit. The precipitated GTP-bound protein samples were subjected to SDS-PAGE, followed by Western blot analysis of Rho expression. Densitometric analysis of the respective blot is shown at the bottom. Bars represent the means ± SD of relative band intensities from independent triplicate samples. For two-group comparisons, a two-sided Student’s *t*-test (** *p* < 0.01) was performed. (**B**) C28/I2 cells were pretreated with simvastatin (Sim; 10 µM) or Y27632 (Y; 10 µM) for 24 h, followed by stimulation with HMW-HA (100 µg/mL) for 1 h. Whole cell lysates were subjected to SDS-PAGE and analyzed by Western blot for phospho-Akt and Akt expression. Equivalent protein loading was confirmed by β-actin levels. Densitometric analyses of the respective blots are shown at the bottom. Bars represent the means ± SD of relative band intensities normalized to Akt from independent triplicate samples. Data were analyzed using Tukey’s test following one-way analysis of variance (ANOVA). ** *p* < 0.01, compared with HMW-HA-treated cells.

**Figure 7 biomedicines-13-00376-f007:**
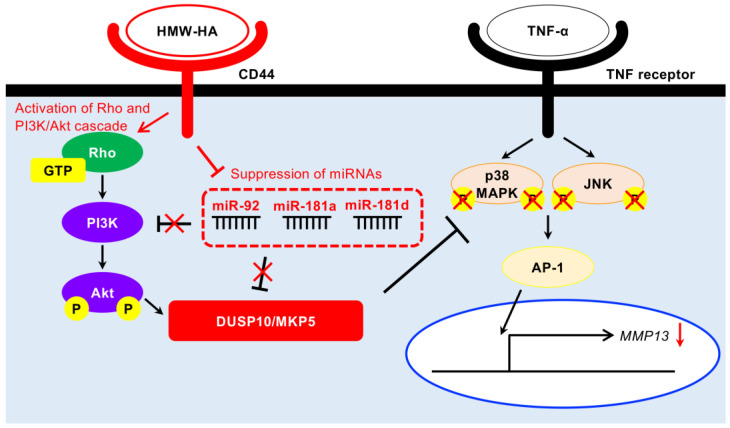
Schematic representation of the molecular mechanisms regulating DUSP10/MKP5 expression induced by HMW-HA in human chondrocytes.

**Table 1 biomedicines-13-00376-t001:** Predicted binding sites for miR-92a-5p, miR-181a-5p, miR-181b-5p, and miR-181d-5p in the 3′- untranslated region (UTR) DUSP10 mRNA.

	Predicted Consequential Pairing of Target Region (Top) and miRNA (Bottom)
Position 449–456 of DUSP10 3′-UTR hsa-miR-92a-5p	5′-…CACAGAAAACAAAAACCCAACCA… | | | | | | | | | | 3′-UCGUAACGUUGGCUA- GGGUUGGA
Position 876–882 of DUSP10 3′-UTR hsa-miR-181a-5p	5′-…UACAUAUGUAUAUCAGAAUGUAA… | | | | | | 3′-UGAGUGGCUGUCGCAACUUACAA
Position 876–882 of DUSP10 3′-UTR hsa-miR-181b-5p	5′-…UACAUAUGUAUAUCAGAAUGUAA… | | | | | | 3′-UGGGUGGCUGUCGUUACUUACAA
Position 876–882 of DUSP10 3′-UTR hsa-miR-181d-5p	5′-…UACAUAUGUAUAUCAGAAUGUAA… | | | | | | 3′-UGGGUGGCUGUUGUUACUUACAA

## Data Availability

The raw data supporting the conclusions of this article will be made available by the authors on request.

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
