# Peer review of "Mechanisms Underlying the Stimulation of DUSP10/MKP5 Expression in Chondrocytes by High Molecular Weight Hyaluronic Acid"

_biomedicines, 2025, doi:10.3390/biomedicines13020376_

Round 1
Reviewer 1 Report
Comments and Suggestions for Authors
-
Expand on the molecular mechanisms involving HMW-HA and DUSP10/MKP5 regulation, providing a clearer differentiation of how each pathway contributes to MMP13 suppression.
-
Include more robust statistical analysis, such as effect sizes or confidence intervals, to better support the findings, especially when comparing multiple groups.
-
Discuss how the findings align or contrast with existing studies on HMW-HA and miRNA regulation, focusing on miRNA-mediated transcriptional regulation in cartilage cells.
-
Highlight the potential clinical implications of the findings more explicitly, particularly for osteoarthritis (OA) and other inflammatory conditions.
-
Acknowledge and address the limitations of the in vitro C28/I2 cell model, suggesting how in vivo models or human clinical data could validate the observed mechanisms.
-
Include correlations between the levels of miR-92a, miR-181a, and miR-181d and DUSP10/MKP5 expression to strengthen claims about their regulatory roles.
-
Enhance the discussion with a schematic model or diagram illustrating the interplay between HMW-HA, CD44, PI3K/Akt, RhoA signaling, and miRNA suppression.
-
Explore whether the mechanisms identified might be relevant to other diseases involving extracellular matrix regulation or inflammatory pathways.
-
Provide specific recommendations for future research, such as exploring additional signaling pathways or testing the therapeutic efficacy of HMW-HA in animal models.
Author Response
We would like to express many thanks for the reviewer’s comments and suggestions which were very helpful to further improve this manuscript. We send a revised manuscript together with the answer to the reviewer’s comments. In the answer to the reviewer’s comments, we highlighted the changes using red font.
Comments 1: Expand on the molecular mechanisms involving HMW-HA and DUSP10/MKP5 regulation, providing a clearer differentiation of how each pathway contributes to MMP13 suppression.
Response 1: We appreciate your helpful suggestions. We modified the Discussion in the revised manuscript (lines 301-306) with some references (Ref. 27-31).
Comments 2: Include more robust statistical analysis, such as effect sizes or confidence intervals, to better support the findings, especially when comparing multiple groups.
Response 2: We appreciate your important suggestions. Confidence intervals describe the uncertainty of the estimate and not the data, so the variance of the variable was described by its standard deviation. To improve clarity, we included dot plots in the graph bars of the revised figures (Fig. 1-6).
Comments 3: Discuss how the findings align or contrast with existing studies on HMW-HA and miRNA regulation, focusing on miRNA-mediated transcriptional regulation in cartilage cells.
Response 3: Thank you for your important comments. There are few research reports on the involvement of miRNAs in the bioactivity of HMW-HA in chondrocytes. However, we did not mention about it. So, we modified the Discussion in the revised manuscript (lines 329-331) with some references (Ref. 43 and 44).
Comments 4: Highlight the potential clinical implications of the findings more explicitly, particularly for osteoarthritis (OA) and other inflammatory conditions.
Response 4: As noted by the reviewer’s we added the potential clinical implications of HMW-HA for OA and modified the Discussion in the revised manuscript (lines 399-401).
Comments 5: Acknowledge and address the limitations of the in vitro C28/I2 cell model, suggesting how in vivo models or human clinical data could validate the observed mechanisms.
Response 5: As noted by the reviewer’s comments, we added some suggestions in the Discussion (lines 396-403) of the revised manuscript with some references (Ref. 64 and 65).
Comments 6: Include correlations between the levels of miR-92a, miR-181a, and miR-181d and DUSP10/MKP5 expression to strengthen claims about their regulatory roles.
Response 6: Thank you for your important suggestions. For further insight into the regulatory roles of each miRNA in on the expression levels of DUSP10/MKP5, the binding of miR-92a, miR-181a and miR-181d to the 3'-UTR of the target gene (DUSP10/MKP5) should be assessed in addition to analysis using bioinformatic tools. However, we did not mention about it. So, we modified Discussion in the revised manuscript (lines 338-340).
Comments 7: Enhance the discussion with a schematic model or diagram illustrating the interplay between HMW-HA, CD44, PI3K/Akt, RhoA signaling, and miRNA suppression.
Response 7: We appreciate your critical suggestions. We modified Figure 7 in the revised manuscript to provide a clear explanation of the molecular mechanisms involved in the suppression of TNF-α-induced MMP13 expression by HMW-HA found in this study.
Comments 8: Explore whether the mechanisms identified might be relevant to other diseases involving extracellular matrix regulation or inflammatory pathways.
Response 8: According to the reviewer’s comments, we modified the Discussion (lines 407-409) of the revised manuscript with some references (Ref. 66 and 67).
Comments 9: Provide specific recommendations for future research, such as exploring additional signaling pathways or testing the therapeutic efficacy of HMW-HA in animal models.
Response 9: Thank you for your comments. We added the possibility of further signaling involvement in the Discussion (lines 391-394) of the revised manuscript with reference (Ref 63).
Reviewer 2 Report
Comments and Suggestions for Authors
Review of biomedicines-3425785: Mechanisms Underlying the Stimulation of DUSP10/MKP5 Expression in Chondrocytes by High Molecular Weight Hyaluronic Acid
This study investigates how high molecular weight hyaluronic acid (HMW-HA) enhances the expression of DUSP10/MKP5 in human chondrocytes, contributing to its chondroprotective effects. HMW-HA activates Akt phosphorylation via CD44 and suppresses miRNAs (miR-92a, miR-181a, miR-181d) that negatively regulate DUSP10/MKP5. Additionally, the study reveals that HMW-HA activates RhoA-ROK signaling, further promoting Akt phosphorylation. These mechanisms—PI3K/Akt, RhoA-ROK signaling, and miRNA regulation—provide valuable insights for potential osteoarthritis treatments.
This manuscript provides valuable insights; however, the experiments were conducted exclusively using the C28/I2 cell line, without including human articular primary chondrocytes or other chondrocyte cell lines, which diminishes the strength and generalizability of the manuscript's results.
Therefore, I have the following comments:
• In the abstract, the authors claim that “in a human articular chondrocyte cell line (C28/I2 cells),”which is incorrect. The C28/I2 cell line is an immortalized chondrocyte line derived from rib cartilage of a 15-year-old female and transduced with simian virus 40 (SV40) containing the large T-antigen [1, 2].
• The abstract also states, “Elucidating these detailed molecular mechanisms may provide insights into novel therapeutic strategies utilizing HMW-HA for osteoarthritis treatment.” Since all experiments were performed using C28/I2 cells, not human primary chondrocytes, this statement is not appropriate. I suggest that the authors perform key experiments using at least three independent human articular chondrocyte preparations.
• To improve clarity, it would be beneficial to include dot plots in the figures (1 through 6).
• In Table 1, the authors show the predicted binding sites for miR-92a-5p, miR-181a-5p, miR-181b-5p, and miR-181d-5p in the 3′-untranslated region (UTR) of DUSP10 mRNA, rather than providing direct binding evidence such as a luciferase assay, which weakens the strength of the manuscript's results.
• In the original images for blots/gels: The Western blotting results are shown in the manuscript, but the authors have not provided data from three independent experiments. Could the authors please provide the images for all three independent experiments?
References:
1. Claassen H, Schicht M, Brandt J, Reuse K, Schädlich R, Goldring MB, Guddat SS, Thate A, Paulsen F. C-28/I2 and T/C-28a2 chondrocytes as well as human primary articular chondrocytes express sex hormone and insulin receptors--Useful cells in study of cartilage metabolism. Ann Anat. 2011 Feb 20;193(1):23-9. doi: 10.1016/j.aanat.2010.09.005. Epub 2010 Oct 25. PMID: 20971625; PMCID: PMC3937963.
2. Goldring MB. Immortalization of human articular chondrocytes for generation of stable, differentiated cell lines. Methods Mol Med. 2004; 100:23-36. doi: 10.1385/1-59259-810-2:023. PMID: 15280585.
Author Response
We would like to express many thanks for the reviewer’s comments and suggestions which were very helpful to further improve this manuscript. We send a revised manuscript together with the answer to the reviewer’s comments. In the answer to the reviewer’s comments, we highlighted the changes using red font.
Comments 1: In the abstract, the authors claim that “in a human articular chondrocyte cell line (C28/I2 cells),” which is incorrect. The C28/I2 cell line is an immortalized chondrocyte line derived from rib cartilage of a 15-year-old female and transduced with simian virus 40 (SV40) containing the large T-antigen[1, 2].
Response 1: Thank you for your suggestions. We modified the abstract (line 20) in the revised manuscript. We also added the references (Ref 21 and 22) provided by the reviewers.
Comments 2: The abstract also states, “Elucidating these detailed molecular mechanisms may provide insights into novel therapeutic strategies utilizing HMW-HA for osteoarthritis treatment.” Since all experiments were performed using C28/I2 cells, not human primary chondrocytes, this statement is not appropriate. I suggest that the authors perform key experiments using at least three independent human articular chondrocyte preparations.
Response 2: We appreciate your critical suggestions. We modified the abstract (lines 36-37) in the revised manuscript. As the reviewer pointed out, validation using human articular chondrocytes, in addition to animal model experiments, is needed to strengthen the results of this study, but we did not mention about it. So, we modified the Discussion in the revised manuscript (lines 401-404).
Comments 3:To improve clarity, it would be beneficial to include dot plots in the figures (1 through 6).
Response 3: According to the reviewer’s suggestions, we included dot plots in the graph bars of the revised figures (Fig. 1-6).
Comments 4: In Table 1, the authors show the predicted binding sites for miR-92a-5p, miR-181a-5p, miR-181b-5p, and miR-181d-5p in the 3′-untranslated region (UTR) of DUSP10 mRNA, rather than providing direct binding evidence such as a luciferase assay, which weakens the strength of the manuscript's results.
Response 4: We appreciate your important comments. As noted by the reviewer, to strengthen the hypotheses obtained from in silico analysis, binding between microRNAs and MKP5/DUSP10 needs to be elucidated. However, we have not mentioned about it. So, we modified Discussion in the revised manuscript (lines 338-340).
Comments 5: In the original images for blots/gels: The Western blotting results are shown in the manuscript, but the authors have not provided data from three independent experiments. Could the authors please provide the images for all three independent experiments?
Response 5: Thank you for your comment. As the presentation of multiple gels complicates and confuses the presentation of results, a typical image of blot is included in each Figure. The western blot analysis was performed a single gel and the results have been cropped and posted. The original images are uploaded as “Supplemental Figure”.
Round 2
Reviewer 2 Report
Comments and Suggestions for Authors
Review of Biomedicines-3425785: Mechanisms Underlying the Stimulation of DUSP10/MKP5 Expression in Chondrocytes by High Molecular Weight Hyaluronic Acid
The experiments in this manuscript were conducted exclusively using the C28/I2 cell line, without including human articular primary chondrocytes or other chondrocyte cell lines, which limits the strength and generalizability of the results. Nevertheless, the manuscript offers valuable insights, particularly regarding the PI3K/Akt, RhoA-ROK signaling, and miRNA regulation mechanisms, which have potential implications for osteoarthritis treatments.
Therefore, this manuscript provides important findings and is recommended for publication in Biomedicines.